# Searching the Dark Genome for Alzheimer’s Disease Risk Variants

**DOI:** 10.3390/brainsci11030332

**Published:** 2021-03-06

**Authors:** Rachel Raybould, Rebecca Sims

**Affiliations:** 1UK Dementia Research Institute in Cardiff, Haydn Ellis Building, Cardiff University, Wales CF24 4HQ, UK; 2Division of Psychological Medicine and Clinical Neuroscience, School of Medicine, Cardiff University, Wales CF24 4HQ, UK

**Keywords:** Alzheimer’s disease, long read technology, dark genome, VNTR

## Abstract

Sporadic Alzheimer’s disease (AD) is a complex genetic disease, and the leading cause of dementia worldwide. Over the past 3 decades, extensive pioneering research has discovered more than 70 common and rare genetic risk variants. These discoveries have contributed massively to our understanding of the pathogenesis of AD but approximately half of the heritability for AD remains unaccounted for. There are regions of the genome that are not assayed by mainstream genotype and sequencing technology. These regions, known as the Dark Genome, often harbour large structural DNA variants that are likely relevant to disease risk. Here, we describe the dark genome and review current technological and bioinformatics advances that will enable researchers to shed light on these hidden regions of the genome. We highlight the potential importance of the hidden genome in complex disease and how these strategies will assist in identifying the missing heritability of AD. Identification of novel protein-coding structural variation that increases risk of AD will open new avenues for translational research and new drug targets that have the potential for clinical benefit to delay or even prevent clinical symptoms of disease.

## 1. Introduction

Alzheimer’s disease (AD) is the leading cause of dementia worldwide. Early family studies identified mutations in *amyloid precursor protein* (*APP*) and *presenilin* (*PSEN*) genes [1,2,3] as the cause of rare, early onset, familial forms of the disease. From this, scientists hypothesized the role of the amyloid cascade in the aetiology of sporadic AD. However, recent failures in clinical trials, based on removing either soluble and/or insoluble Aβ or targeting enzymes responsible for cleavage of APP, have thrown doubt on the hypothesis [4,5]. Familial rare mutations, that induce early onset AD (EOAD), constitute less than 1% of all AD cases. Common forms of late-onset AD (LOAD) have heritability estimates of 56–79% [6] and are contributed to by multiple genetic risk factors [7].

Over the past three decades great advances in gene discovery have been made by worldwide collaborative projects such as European AD DNA Biobank (EADB), AD Sequencing Project (ADSP), AD European Sequencing Consortium (ADES) and the International Genomics of Alzheimer’s Project (IGAP). These pioneering studies are summarized elsewhere [7]. To date around 70 genetic variants have been reported to be associated with AD risk (Figure 1) and these discoveries have indicated the involvement of pathways, additional to APP metabolism, such as immunity, ubiquitination, endocytosis, lipid metabolism and tau binding [8,9]. Furthermore, generation of large databases of genetic data have led to discoveries of genetic prediction strategies such as the polygenic risk score for AD [10].

Ultimately, the genetic research carried out to date has contributed vastly to our understanding of Alzheimer’s disease and gives hope that AD can be diagnosed and treated before the onset of irreversible neurodegeneration. However, it is estimated that a substantial proportion (approximately half) [12,13] of the genetic variance of LOAD is not yet accounted for. The majority of genetic research has focused on single nucleotide polymorphisms (SNPs), and utilized genome-wide association studies (GWAS), due the ease of genotyping in a high-throughput, cost and time effective manner. GWAS uses genotyping arrays based on known genetic variants. Therefore, the undiscovered genetic heritability is likely to be accounted for by other forms of variation, especially rare protein-coding or large variable number of tandem repeats (VNTRs) that either increase the risk of disease or offer protection against it. Next generation sequencing (NGS) has enabled researchers to screen large sample populations to detect novel rare small nucleotide variants for AD however, short read length limits the detection of larger structural variants within the DNA. Copy number variants (CNVs), inversions and translocations and VNTRs, in particular, have been unintentionally overlooked because of limited technological advances. This review highlights the importance of searching the genome for large structural variants that may confer risk or protection for AD and reviews current technological and analytical advances and limitations.

## 2. Next Generation Sequencing in AD Studies

Next Generation Sequencing (NGS) is generally based on spatially separated amplified DNA regions that cover specific sequences of genome such as the whole genome, whole exome or targeted gene regions. In short read sequencing experiments these amplified regions, known as reads, are 75 to 150 base pairs (bp) in length. The amplified regions are sequenced in a flow cell and aligned to a reference genome to detect DNA variation within the sample. The use of short sequences enables DNA sequencing at high throughput and at a high depth in order to detect mutations with high accuracy for a large sample size. The reduction in cost and increase in throughput has led to new discoveries in the field of AD genetics (Table 1). For example, in families with an autosomal dominance inheritance pattern of AD without PSEN or APP mutations, whole exome sequencing (WES) led to the discovery of novel *Sortilin related receptor* (*SORL1*) mutations. SORL1 is a protein involved in the control of amyloid production and there may be many mutations within this gene that contribute to AD with varying risk based on the damage the variant makes to the SORL1 protein and the rareness of the variant [14]. Recently, WES screening of a large AD case control sample confirmed that carrying *SORL1* rare, damaging protein variants approximately triples the risk of EOAD and doubles the risk of LOAD [15]. WES has also uncovered novel damaging variants in *TREM2* [16,17], *IGHG3* and *ZNF655* [18] and *NSF* [19]. The encoded proteins are functional in pathways known to confer AD risk such amyloid processing or inflammation.

Whilst NGS technology has led to discovery of variants that increase risk of AD, NGS has also identified variants that offer protection against AD in *APP*, *CBX3* and *PRSS3* [15,20]. Identification of protective variation in AD is as important as identifying disease causing variation. These protective variants highlight biological pathways that provide resilience to disease and have the potential for therapeutic intervention for improved treatment or even prevention of clinical AD.

**Table 1 brainsci-11-00332-t001:** A summary of NGS short read sequencing studies in AD. Association statistically significant at *p* < 10^−6^ exome wide for WES and *p* < 10^−8^ genome-wide for Whole Genome Sequencing (WGS).

Author	Method	Cohort	Cohort Size (n)	Outcome
[21]	Agilent SureSelect Human & Illumina sequencing	Autosomal dominant EOAD without known mutations	14	7 unknown mutations detected in SORL1 gene.
[20]	WGS, Illumina bead chip, and imputation	Icelandic	1795 WGS 71743 chip	Coding mutation A673T in APP protects against AD.
[16]	WGS, Illumina bead chip, and imputation	Icelandic	1795 WGS	rs75932628-T (R47H) of TREM2 increases risk of LOAD.
[17]	Illumina WES TruSeq, Illumina Sequenced	Caucasian from UK, US and Canada	1092 AD cases 1107 Controls	rs75932628-T (R47H) of TREM2 increases risk of LOAD.
[14]	Roche NimbleGen v2 and v3, SeqCap^®^ EZ Exome, Illumina TruSeq	1908 Dutch AD cases and Controls	640 cases 1268 Controls	181 unique SORL1 variants detected. Developed a strategy to classify SORL1 variants into five subtypes ranging from pathogenic to benign.
[22]	Targeted sequencing using Agilent HaloPLex™ and Illumina sequencing	Unrelated LOAD and matched controls	772 LOAD 757 Controls	Intronic variant rs78117248 in ABCA7 showed strongest association with AD. Loss of function mutations may be a potential pathogenic mechanism.
[18]	WGS and WES data generated by ADSP using Roche NimbleGen and Illumina Rapid Capture Exome	European-American and Hispanic	172 AD 171 Controls	Novel variants in previously reported AD risk genes and variants in novel genes IGHG3 and ZNF655 were detected.
[19]	Agilent SureSelect and ADSP using Roche NimbleGen and Illumina Rapid Capture Exome	Caucasian non APOE*4 carriers from PITT-ADRC ^1^ ADSP ^2^	PITT-ADRC^1^372 AD 348 Controls ADSP^2^ 2113 AD 5139 Controls	Association at novel variant in NSF gene and known loci in TREM2, TOMM40 and APOE was detected.
[15]	Roche NimbleGen, Agilent NextEra, Truseq, HaloPlex, SureSelect, Illumina Nextera^®^ Rapid Capture Exome	25982 Caucasian	12652 AD 8693 controls	Detected protein damaging variants in TREM2, SORL1 and ABCA7 and protective variants in CBX3 and PRSS3.
[23]	ADSP using Nimblegen NimbleGen v2 and v3 and Illumina Nextera^®^ Rapid Capture Exome	5142	4889	Identified a VNTR within MUC6 associated with pTau burden.

^1^: University of Pittsburgh Alzheimer’s Disease Research Center (ADRC). ^2^: Alzheimer’s Disease Sequencing Project (ADSP).

## 3. The Dark Genome

One of the key limitations of recent discoveries by short read NGS is that not all generated sequences are adequately mapped or aligned to the genome. This can be due to technical issues arising from poor quality of DNA or type of kit used. It is also due to the nature of the human genome. The human genome consists of many repeat elements and gene duplications and translocations making it too complex to map short read sequences with certainty (Figure 2). There are also camouflaged regions where reads cannot be mapped because the region has been duplicated in the genome. This missing element is described as the “dark genome”.

A systematic analysis of dark regions in short read NGS data was undertaken for ten unrelated males from the ADSP [24]. The study reported that standard short-read sequencing leaves 36,794 dark regions across 6054 gene bodies, including protein coding exons from 748 genes. Sixty-two percent of the dark gene bodies are dark because of mapping quality and 71.1% of gene regions were replicated three or more times within the genome leading to poor mapping quality. When they applied long read technologies from 10x Genomics, PacBio and Oxford Nanopore Technology they reported a reduction in dark protein-coding regions to approximately 50.5%, 35.6%, and 9.6% for each technology, respectively.

## 4. The importance of Large Structural Variants in Complex Disease

Large structural variants arise from errors in DNA repair, recombination, translocations and result in DNA variants for example, deletions, duplications, insertions and translocations. Repeats are thought to constitute approximately 3% of the human genome [25]. Short tandem repeats (STRs) are the most commonly observed variety of repeat in the human genome. STRs such as homonucleotide, dinucleotide and trinucleotide repeats have been implicated in regulating gene expression through mechanisms such as altering splice sites, modulating binding of transcription factors and changing promoter region DNA sequences [26]. Furthermore, larger, more complex repeats for instance variable number of tandem repeats (VNTRs) and copy number variants (CNVs) are not as common as STRs but contribute to rare forms of disease with high pathogenic effect.

The 1000 Genomes project and disease sequencing studies have generated maps that allow scientists to understand how VNTRs are associated with a specific type of disease. There are many published Mendelian VNTRs and regions of genomic instability that are associated with distinct phenotypes. Deletions on 22q11 are the most common form of microdeletions in humans. The deletions on 22q11 are thought be caused by large chromosome specific low copy repeats that affect chromosomal events during meiosis [27] and result in complex phenotypes including DiGeorge Syndrome and Velo Cardio Facial syndrome (VCF). A “CGG” trinucleotide repeat within the *Fragile X Mental Retardation 1* (*FMR1)* gene, on the X chromosome, can result in hyper-methylation and silencing of FMR1. Those who possess 55 or more copies of the repeat within the *FMR1* gene tend to have a complex disorder that features intellectual disabilities, psychiatric disabilities, neuro-degenerative disorders combined with facial abnormalities. Huntington’s disease (HD) is caused by a “CAG” trinucleotide repeat within exon 1 of the *Huntington* (*HTT*) gene. HD severity increases with increasing CAG repeat length repeat length and expansions of greater than 36xCAG result in complex phenotypes including degeneration of the cortex and striatum. Furthermore, HTT CAG repeat length only accounts for up to half of the variation in age at onset [28] and disease modifiers such as FAN1 influence DNA repair and CAG expansion [29].

Uncovering repeats associated with complex disease is particularly difficult due to many genes and environmental factors synchronizing together to create a phenotype. In addition, sub-phenotypes cause heterogeneity within a disease and contribute to differing ages at onset and symptom profiles. Nucleotide repeats associated with complex phenotypes often coincide with Mendelian disease loci [30]. A study that generated a comprehensive map of 11,700 CNVs, identified 30 trait associated SNPs that were in high linkage disequilibrium with CNVs [31]. It is highly plausible that some AD genome-wide significant (GWS) hits, may be in LD with nearby CNVs or other structural variation.

### Structural Variation in Dementia

Candidate gene approaches have identified a proportion of large structural DNA variation in AD (Table 2). However, these studies commonly had small sample sizes, lacked power and regularly failed to replicate. Meaning a number of associations documented in the literature warrant further investigation.

The *Insulin* (*INS*) gene possesses a tandem repeat of 15 bp which ranges from 400 bp to 8000 bp in size [32]. This VNTR was genotyped in a Caucasian population but no association with AD was reported. However, this group reported an earlier age at onset (AAO) in participants homozygous for Class III INS VNTR repeat [33] which contains 139 repeats [32].

Interleukin 6 (IL6) possesses an AT rich region, spanning over 500 bp, with 4 known alleles: A, B, C and D [34]. The D allele of the IL6 VNTR has been reported to be associated with increased risk of AD [35] and the C allele is reported to offer protection against AD and delay the onset of symptoms [35,36]. Subsequent to these publications, the role of inflammation in AD has been reported through GWAS and pathway analysis but association at IL6 has not yet been reported at the genome-wide significance level.

The *Monoamine oxidase inhibitor A* gene (*MAOA*) is located within a region of known instability on the X chromosome and plays a wide range of roles in circadian rhythm and degradation of neurotransmitters. Within the promoter region of *MAOA* is a 30 bp VNTR that influences the expression of MAOA. Association within the *MAOA* locus has not been reported by GWAS or WES studies. However, the VNTR within the promoter region of *MAOA* has been reported to play a in sleep disturbances in AD [37] and cognitive function in AD [38].

Nitric oxide is a potential neurotoxin and is produced by three isozymes of nitric oxide synthase (NOS). The link between AD and NOS is not clear and NOS may have both neuroprotective and neurotoxic effects [39]. *NOS1* gene possesses a CA dinucleotide VNTR located in the promoter and is classified as long (more than 10 repeats) or short (10 repeats or less) in association studies. The *NOS1* promoter VNTR has been reported to be associated with increased AD risk and interacts with the largest single genetic risk factor for AD, *APOE* ε4 allele to further increase AD risk [40].

Pathogenic repeats have also been reported to contribute to other neuro-degenerative diseases. For example, in frontal temporal dementia (FTD) and amyotrophic lateral sclerosis (ALS), a hexamer (GGCCC) repeat in *C9orf72*, has accounted for both familial and sporadic forms of disease [41,42,43]. The repeat, over 30 repeats in length, stretches over 180 bp and can contain insertions and deletions in the flanking regions making it difficult to sequence using short read technology and difficult to genotype using repeat primed PCR or Southern blot. The repeat expansion in *C9orf72* may lead to an alternatively spliced transcript and formation of nuclear RNA foci. The repeat in *C9orf72* was originally detected in ALS and FTD cohorts using Sanger sequencing [41] and recently Expansion Hunter (detailed below) was developed to enable genotyping by short length read technology [44].

**Table 2 brainsci-11-00332-t002:** A summary of published associations of large structural variation with AD.

Author	Gene	Polymorphism Size	N Cases	N Controls	Outcome
[36]	IL6	500–800 bp AT rich repeat with 4 alleles (A,B,C or D)	102	191	C allele is associated with AD protection (*p* = 0.025) and with delayed onset of AD (*p* = 0.034).
[33]	INS	400–8000 bp	58	161	No association with AD (*p* = 0.873). Class III INS VNTR homozygotes had earlier AAO (*p* = 0.0002).
[35]	IL6	500–800 bp AT rich repeat with 4 alleles (A,B,C or D)	184	273	C allele was negatively associated with AD (*p* = 0.001), DD genotype associated with AD (*p* = 0.015; OR = 1.636, 95%CI = 1.101-2.432). Patients with D allele of VNTR had higher plasma levels of IL6 (*p* = 0.001).
[37]	MAOA	30 bp repeat. Alleles with 3, 3.5, 4 and 5 repeats	425	n/a	4 repeats associated with sleep disturbance in AD patients (*p* = 0.005)
[45]	MAOA	30 bp repeat. Alleles with 3, 3.5, 4 and 5 repeats	44	63	3.5 and 4 repeats associated with increased expression of MAOA in AD pineal glands.
[38]	MAOA	30 bp repeat. Alleles with 3, 3.5, 4 and 5 repeats	193	n/a	Sex- genotype interaction may determine cognitive scores
[40]	NOS1	CA repeat Long >10 repeats Short 10 repeats	184	144	The short allele of NOS1 VNTR is associated with AD (*p* = 0.0009) and interact with APOEε4 allele to increase risk.
[46]	ABCA7	25 bp repeat 12–247 repeats			An increase in VNTR length was associated with decreased ABCA7 expression. Increased VNTR length was associated with alternative splicing.
[23]	MUC6	VNTR exon 31 Length 9–12 kbp	119 Cohort 173 Replication	n/a	pTau pathology associated with increased VNTR length (*p* = 0.031)

## 5. Shedding Light on the Dark Genome

As mentioned above, short read technologies are unable to detect large structural variants due to poor mapping quality (Figure 2). Current technology and bioinformatics are changing to enable us to fill in the heritability gaps created by the dark genome. More detailed analysis of short read data could potentially lead to detection of AD risk or protective variants hidden within dark regions of data.

### 5.1. Generic Molecular Biology Approaches

Where NGS has identified regions that poorly map to the genome, some studies have applied “traditional” molecular biology methods such as Sanger sequencing, long range PCR and Southern blot to further characterize genomic locations indicative of association with AD risk:

Katsumata et al. [23] used SKAT-O to analyse short read WES data, generated by ADSP, to identify novel rare risk variants for LOAD. SKAT-O detected both positively and negatively associated variants within the *Mucin6* gene (*MUC6*) however, association at these variants did not pass quality control due to poor sequence mapping to the human genome. The variants within *MUC6* were within a region with complex tandem repeats and further inspection, using long range PCR, cloning, Sanger sequencing and restriction digestion identified a highly polymorphic VNTR within *MUC6*. This group then explored the relationship between AD neuropathology in autopsied individuals and reported that individuals with longer VNTR regions had significantly more pTau burden.

There is a similar story for the *ATP-Binding Cassette Subfamily A Member 7* gene (*ABCA7*). Premature termination codons (PTCs) are 4 to 5 times more prevalent in *ABCA7* in AD patients however, until recently, the functional variant that accounted for the GWAS signal remained undetected. Targeted gene sequencing, using short read technology, detected association at PTCs [22] but these still did not explain the GWAS association. Sanger sequencing of a stretch of DNA up to 3 kbp identified a 25 bp repeat within the splice donor site of exon 18 in *ABCA7*. Subsequent genotyping using Southern blot showed that individuals carrying an allele > 5720 bp (“expanded”), were mostly AD patients [OR 4.5 (95% CI 1.3–24.2), *p* = 0.008 [46].

Methods such as long-range PCR, Sanger sequencing and Southern blot are successful in characterizing and genotyping long stretches of DNA harbouring complex repeat elements. Unfortunately, these methods are not high throughput or time efficient and require large quantities of un-degraded DNA.

### 5.2. Bioinformatics Approaches

A number of studies have attempted to investigate VNTRs by applying novel bioinformatics approaches to short read NGS data. Ebbert et al. [24] developed an algorithm to extract all regions within gene bodies with a mapping quality (MAPQ) score less than 10 and then identified the regions using BLAT [47]. This algorithm was applied to the ADSP data in an attempt elucidate most of these dark regions. They identified a frameshift deletion in C3b and C4b binding region of known AD risk gene *Complement Receptor 1* (*CR1*) but further work is required to establish this association. Ebbert et al. [24] also identified dark regions within additional gene bodies, linked with AD association including *ABCA7*, *INPP5D*, *IQCK* and *HLA*.

Recursive exact matching is a method that can detect large regions of repeats within sequence data [48]. The REVEAL algorithm utilizes recursive exact matching and creates a hierarchal tree of decisions based on maximally unique matching (MUM). The REVEAL algorithm gives a high resolution, enables alignments of repetitive sequence data and can detect inversions and translocations [48].

Imputation tools have been developed that can combine datasets taken from NGS and SNP genotyping panels to predict the genotype of short tandem repeats (STRs). Saini et al. [49] generated a catalogue of STR genotypes from a cohort of families using NGS (150 bp read length). They then imputed the STR genotypes into 1000 Genomes SNP data [50] to create a SNP+STR haplotype reference panel. The reference panel generated can be used to impute STRs into larger genetic datasets such as AD cohorts where SNP genotype data exist for tens of thousands of samples. The haplotype panel created by Saini et al. (2018) is limited in that it can only accurately predict STRs that are fully spanned by a read. It cannot detect or predict long and more complex repeats due to the read length not spanning the full length of the repeat.

Tang et al. [51] developed TREDPARSE, to identify each allele length at predefined STR loci by using Illumina WGS sequence data (read length 150 bp) that are sampled at sufficient depth. TREDPARSE is based on 4 models of calling STRs: model 1 is based on a read that spans the entire repeat and flanking region; model 2 is based on partial reads of the repeat including flanking regions; model 3 is based on reads that consist of repeat only; model 4 is based on paired end reads where one read flanks the beginning of a repeat and the paired read flanks the end of a repeat. These 4 modes allow the allele calling of both long and short repeats on short read NGS data.

Expansion Hunter is a tool that can be used to detect both long and short repeats [44]. Expansion Hunter catalogues repeats associated with the repeat of interest and approximates the size of the repeat based on the flanking reads. A maximum likelihood genotype is created by the spanning, flanking and in-repeat reads. Expansion Hunter works well for repeats because it relies on the basis that all repeats are identical. Expansion Hunter does not work as well for VNTRs due to the complex variation between repeat units.

adVNTR [52] was developed for genotyping VNTRs at targeted loci in a donor genome. For any target VNTR in a donor, adVNTR reports an estimate of repeat unit counts and point mutations within the repeat units. Hidden Markov Models are trained using reference assemblies as well as training of the flanking regions of VNTRs in the reference genome.

The above approaches can be applied to currently existing AD genotype and sequencing datasets to explore the genome for additional structural variants that contribute to the heritability of AD. However, even with these improvements in bioinformatics methodology there still remain large structural variants that are too large to detect using short read NGS technology.

### 5.3. Long Read Sequencing Technologies

Long read technologies offer a means to examine the genome with high resolution with read lengths up to 80 kbp or larger. Long read sequencing is slowly becoming more affordable and higher throughput. Therefore, long read sequencing is becoming a more favourable method for detecting and genotyping large structural variants. There are three main providers for long read sequencing supplying to the market. These are assessed in detail elsewhere [53] and reviewed in brief here.

The SMRT Sequel II is a recent release from PacBio. The Sequel II can sequence DNA strands over 15 kilobases in length. SMRT Sequel II utilizes SMRTbell technology, which is created by ligating hairpin adaptors to both ends of a target double stranded DNA molecule [54]. A polymerase is used to replicate the full length of the DNA strand allowing many copies of the full-length strand to be replicated. Multiple sequenced copies of each strand allow higher accuracy of nucleotide calling and reduces errors in reads. It is possible to multiplex samples using adapter primers reducing cost of sequencing. This technology is ideal for detecting detect structural variants (SVs), CNVs, and large indels ranging in size from tens to thousands of base pairs.

Oxford Nanopore Technologies (ONT) offer a number of platforms for long read sequencing: The MinION, GridION and PromethION can produce extremely long reads up to 4 Mbp in length. The ONT platforms perform sequencing of a DNA strand based on ionic changes in current within a flow cell. DNA, with DNA-protein adapters tightly bound to a polymerase, is placed in a flow cell with ionic solutions. The polymerase allows stepwise unwinding and movement of the DNA strand through a pore and ionic current changes are measured as the DNA strand passes through. The ionic current changes are used to construct a “squiggle plot” that is processed by minKNOW software into DNA sequence data. The MinION is a small, portable device and is suitable for work in the field. The GridION and PromethION are designed for larger scale, high-throughput projects. Whilst ONT sequencers can sequence very long strands of DNA, the DNA reads are based on up to two copies of the strand. Therefore, this method of sequencing can have a high error rate of base calling.

An alternative approach to long read whole genome sequencing is offered by 10X Genomics. The Chromium™ Genome Protocol generates long-range information across the length of individual DNA molecules with a mean length greater than 65 kbp. Using a microfluidics chip, single strands of template genomic DNA are encapsulated in separate Gel Bead-In-Emulsions (GEMs) containing copies of a unique barcoded primer. Once the DNA strands are partitioned within a GEM and labelled with unique barcodes specific for that GEM, the fragments are sheared and sequenced using classical Illumina short read technology. The weakness of this approach is that it utilizes PCR and increases the risk of PCR based errors. Therefore, deep sequencing of greater than 30x copies is required.

Ultimately, what determines the success of long read sequencing experiments, is DNA quality. Intact, un-degraded and pure DNA is required for these methods. Ideally, fresh DNA that has not been subjected to freeze-thaw or fixation is a prerequisite. Currently, large DNA banks possess tens of thousands of DNA samples collected over the past 15 years or more and many of these samples would be deemed unsuitable for long read sequencing due to DNA degradation or impurity.

## 6. Conclusions

Candidate gene studies have identified large structural variation associated with AD, but these have not been systematically or robustly evaluated. Large and small tandem repeats play a role in regulating gene expression but have been overlooked in complex disease due to limited technological advances.

Approximately 50% of the heritability for AD is unaccounted for and the search for variants within dark regions of the genome is highly likely to identify functional structural variation that can be easily modelled and targeted for pharmaceutical intervention.

Recent studies have begun to explore the dark genome and reported large structural variants within known AD risk genes *CR1* and *ABCA7*. There are also regions within *INPP5D*, *IQCK* and *HLA*, as well as valid AD candidate genes, which contain dark areas not assayed by genome-wide genotyping or short read sequencing technologies. These regions require further investigation. The studies, to date, have solely focused on data from the ADSP. There are now many more WES and whole genome datasets (e.g., ADES) that can be interrogated with novel bioinformatics approaches for disease associated structural variation within dark regions of the genome.

While *in silico* bioinformatics approaches will aid the discovery of novel large repeat variants within the dark regions of the genome, the gold standard approach will be long read sequencing that does not rely on reference panels and potentially imprecise imputation. At this time long read sequencing technology remains prohibitively expensive to undertake large scale case–control analyses. More sophisticated study designs are warranted. However, given time, the cost will reduce, allowing the field to robustly and systematically search the dark genome to further understand the pathogenesis of AD and allow for the development of treatment that will slow or prevent the progression to disease.

## Figures and Tables

**Figure 1 brainsci-11-00332-f001:**
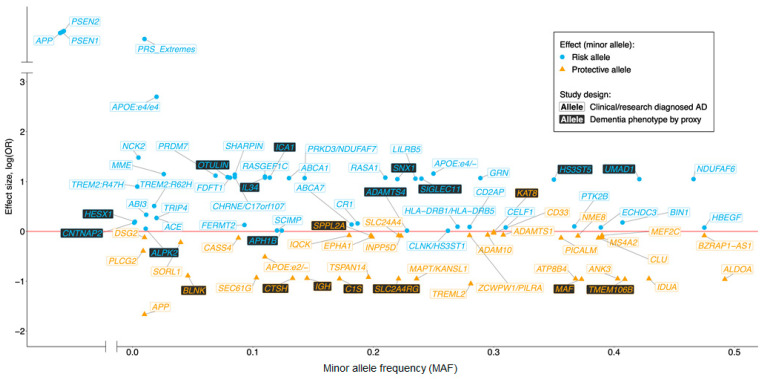
Schematic representation of Mendelian disease-causing genes and loci reaching genome-wide significance (GWS) for single variant (not gene-wide) association with sporadic Alzheimer’s disease (AD). Filled blue and orange points represent risk and protective association, respectively, in AD diagnosed cohorts. Adapted from Sims et al. [7] with the addition of variants identified by EADB [11].

**Figure 2 brainsci-11-00332-f002:**
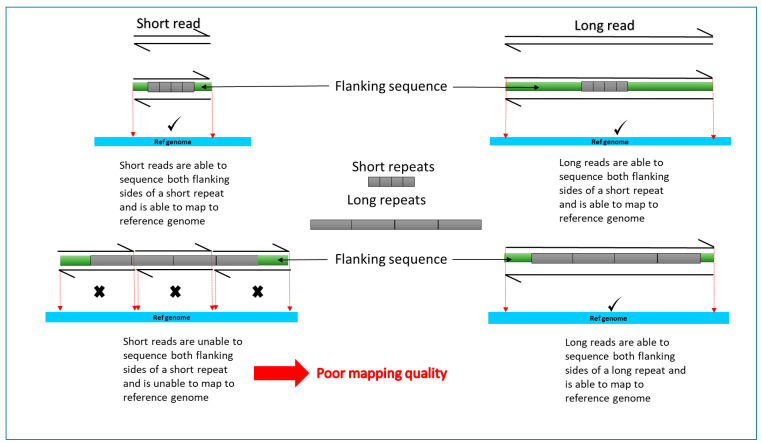
A simplified, schematic diagram comparing short read and long read mapping to reference genome for long and short repeats. Sequencing reads are indicated by partial arrows. Grey rectangles represent target sequence. Flanking sequence is represented by green rectangles. Reference genome used to align the reads is indicated in blue. Reads poorly or not aligned to the reference genome are indicated by X. Success in mapping to reference genome indicated by a tick.

## Data Availability

Not applicable.

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
