# Peer review of "Searching the Dark Genome for Alzheimer’s Disease Risk Variants"

_brainsci, 2021, doi:10.3390/brainsci11030332_

Round 1

Reviewer 1 Report

There are growing interests to better understand the contribution of the “dark” genome, those non-coding sequences, to development and diseases. While most review articles on Alzheimer’s disease focus on known mutations, this manuscript has chosen to examine the unaccounted mutations that likely to reside within the “dark” genome.   

To readers who are new to this topic (dark genome) and on the area of short read sequencing, the authors have provided good illustrations and examples from recent studies to depict the challenges of deciphering these DNA sequence within the “dark genome”. 

In short, this is a balanced review which examine the growing contribution of the non-coding sequence to the pathogenesis of Alzheimer’s disease. 

Author Response

We thank you for taking the time to assess our work. We are delighted that the reviewer saw the importance of the issue we aimed to highlight and considered our work to be thorough and easy to understand. 

The reviewer did not make any comments for us to address in the document.

Reviewer 2 Report

The work presents a comprehensive review of the state of the art in this field. Furthermore, it presents a very interesting approach related to heredity in neurodegenerative diseases. The study of the dark genome is undoubtedly important to know all the genetic contribution to these diseases. Only some minor considerations should be adressed before its publication.

-The manuscript pages are wrongly numbered.

-Figure 1: frequency intead of arequency

-Table 1A. Please indicate the gene in positions 2 (A673T, probably APP), 3 and 4 (R47H, probably TREM2)

-begining page 5 of 13 (2 of 13 in the manuscript): there is some rare text over a line (Illumina Nextera Rapid-Capture Exome)

-line 185 (C9orf72): the lack of splicing is only a possible explanation of the molecular pathogenicity of these expansion. There are other plausible theories (for instance the RAN translation of toxic dipeptides).

-table 2. It seems more appropriate to put all MAOA cases together.

-line 293 p10/13 (7/13 in the manuscript). There are other devices from Nanopore besides MinION (Grid and Prometh), and the longest sequence achieved to date is 2.3 Mb (not 40).

Author Response

We thank the reviewer for taking the time to assess our work. We are delighted that the reviewer saw the importance of the issue we aimed to highlight and considered our work to be thorough and easy to understand, suggesting only minor amendments to the piece. We have addressed the reviewer’s comments/suggestions point-by-point below

  1. The manuscript pages are wrongly numbered.

We apologize, it appears that there was an issue with the template. We have amended the page numbers accordingly for pages 1 to 15.

2 . Figure 1: frequency instead of arequency

We have corrected the spelling of frequency in Figure 1.

3. Table 1A. Please indicate the gene in positions 2 (A673T, probably APP), 3 and 4 (R47H, probably TREM2).

We have entered the relative gene names (APP and TREM2) at positions 2, 3 and 4 in table 1.

4. Begining page 5 of 13 (2 of 13 in the manuscript): there is some rare text over a line (Illumina Nextera Rapid-Capture Exome)

Apologies we don’t see an issue in our document. Perhaps this is a formatting problem with the document received?

5. line 185 (C9orf72): the lack of splicing is only a possible explanation of the molecular pathogenicity of these expansion. There are other plausible theories (for instance the RAN translation of toxic dipeptides).

We agree and have amended the sentence at line 189 to 190 to “The repeat expansion in C9orf72 may lead to an alternatively spliced transcript and formation of nuclear RNA foci.”

6. table 2. It seems more appropriate to put all MAOA cases together.

We agree and have now placed all MAOA cases together in table 2.

7. line 293 p10/13 (7/13 in the manuscript). There are other devices from Nanopore besides MinION (Grid and Prometh), and the longest sequence achieved to date is 2.3 Mb (not 40).

Thank you for highlighting this omission. We have added the GridION and PromethION into the paragraph. We have also amended the longest sequence to “up to 4Mbp” as this is the length ONT claim their sequencers can achieve. The amended  paragraph, lines 293 to 305 now reads “Oxford Nanopore Technologies (ONT) offer a number of platforms for long read sequencing: The MinION, GridION and PromethION can produce extremely long reads up to 4Mbp in length. The ONT platforms perform sequencing of a DNA strand based on ionic changes in current within a flow cell. DNA, with DNA-protein adapters tightly bound to a polymerase, is placed in a flow cell with ionic solutions. The polymerase allows stepwise unwinding and movement of the DNA strand through a pore and ionic current changes are measured as the DNA strand passes through. The ionic current changes are used to construct a “squiggle plot” that is processed by minKNOW software into DNA sequence data. The MinION is a small, portable device and is suitable for work in the field. The GridION and PromethION are designed for larger scale, high-throughput projects. Whilst ONT sequencers can sequence very long strands of DNA, the DNA reads are based on up to two copies of the strand. Therefore, this method of sequencing can have a high error rate of base calling.”